# Centralized Intake Models and Recommendations for Their Use in Non-Acute Mental Health Services: A Scoping Review

**DOI:** 10.3390/ijerph20095747

**Published:** 2023-05-08

**Authors:** Anton Isaacs, Alistair Bonsey, Danielle Couch

**Affiliations:** 1School of Rural Health, Faculty of Medicine Nursing and Health Sciences, Monash University, Warragul, VIC 3820, Australia; 2Victorian and Tasmanian Primary Health Network Alliance, Parkville, VIC 3052, Australia

**Keywords:** health services accessibility, community mental health services, community health services, centralized intake, single entry model, scoping review

## Abstract

Centralized intake [CI] or single-entry models are utilized in health systems to facilitate service access by reducing waiting times. This scoping review aims to consolidate the Literature on CI service models to identify their characteristics and rationales for their use, as well as contexts in which they are used and challenges and benefits in implementing them. The review also aims to offer some lessons learned from the Literature and to make recommendations for its implementation in non-acute mental health services. The findings show that CI is mostly considered when there is increased demand for services and clients are required to navigate multiple services that operate individually. Successful models have meaningfully engaged all stakeholders from the outset and the telephone is the most common mode of intake. Recommendations are made for planning and preparation, for elements of the model, and for setting up the service network. When successfully implemented, CI has been shown to improve access and increase demand for services. However, if CI is not supported by a network of service providers who offer care that is acceptable to clients, the purpose of its implementation could be lost.

## 1. Introduction

When the COVID-19 pandemic swept across the world in the year 2020, the state of Victoria in Australia went into a 111-day lockdown. This resulted in a surge in the number of people needing mental health care [1]. Typically, persons requiring mental health care are referred to public mental health services or private psychologists by their general practitioner. In response to the pandemic, in Victoria a centralized intake model called Head to Help (later renamed Head to Health) was rapidly rolled out to run in parallel to the existing mental health service system in order to improve access to non-acute mental health services [2]. Persons needing mental health care could call a toll-free number which would be answered by a mental health professional who would work with them to find the best ways to get the help they need [2]. This was a relatively new population level service model implemented in the state that was specifically aimed at improving access to care for people with non-acute mental health challenges. 

In late 2022, following the relative success of the Head to Health program, discussions were underway to improve the program. Together with evaluations of the program, it was decided that lessons could also be learned from a review of similar single-entry models trialed around the world in the field of mental health. 

Centralized intake [CI] or single-entry models [SEMs] are utilized in health systems to facilitate service access by reducing waiting times [3]. This is achieved by assembling all clients into a single queue before screening and referring them to the right kind of service or healthcare professional. The process of CI appears to have been first implemented in 1975 in Omaha, USA [4]. Levie and colleagues explain that there was much frustration and misunderstanding among staff of agencies that worked under the umbrella of the Metro Interagency Drug Abuse Program [MIDAP] in Omaha. A CI office was therefore established to improve working relationships between the different drug treatment programs and to make the most appropriate treatment option immediately available to clients [4]. Since then, CI has been adopted in different settings for various reasons. For instance, in the US, CI was introduced to improve substance use programs when existing services were functioning disparately and clients’ treatment needs were not well matched to services resulting in inconsistent service quality, limited access to treatment, and poor service coordination [5]. Similarly, in Australia, CI was trialed to address the difficulties encountered by individuals with severe mental illness who were falling through the gaps in the system owing to barriers in their ability to access services for their multiple and complex needs [6]. In Canada, large disparities in the waiting time for hip or knee surgery across orthopedic surgeons were causing clients’ health to worsen and causing them to lose confidence in the system. CI was therefore introduced to improve patient access by referring them to the next available surgeon [7]. 

CI has also been utilized in various other settings such as home health integrated care [8], physiotherapy for children with complex needs [9], maternal and child health home visiting programs [10], and Lung cancer diagnosis and treatment [11], and has been reported to be a useful model for sub-acute care where there are a number of healthcare professionals or facilities providing similar services [12,13]. Milakovic and colleagues undertook a systematic review of the effects of a CI model on access to specialist physicians and allied health professionals [3]. They found that CI models were associated with a reduction in waiting time from referral to primary care consultation and that such models showed promise to improve access to a range of health services despite the studies having a high risk of bias [3].

However, there continues to be no single definition for this model of care [14] and, although it has shown promise in multiple settings, the Literature at no point consolidates the various ways in which the model can be used as well as its benefits and challenges. A review of the Literature on CI models would be useful for researchers and, more importantly, for policy makers.

## 2. Materials and Methods

A scoping review was conducted to identify and collate the Literature on CI. The purpose of a scoping review is to identify knowledge gaps, scope the body of the Literature or clarify concepts [15]. As opposed to a systematic review, which is typically undertaken to synthesize the evidence from multiple studies on a well-researched topic, a scoping review aims to consolidate the available evidence on a topic area that is emerging [16]. Academic works on CI models are few and far between. Hence, a scoping review was considered the best method of reviewing the topic. The methodological framework of Arksey and O’Malley [17] was adopted to undertake the review. Since its introduction, this framework has been widely used and recognized as useful for conducting scoping reviews [18]. As part of the methodology, the authors suggest an optional stage of consultation with experts. However, a recent report has indicated that there are potential issues such as stakeholder power imbalances during consultations, and therefore this aspect of the method needs further research and clarification [19]. Nonetheless, in this instance, the Project Control Group of the Head to Health program are collectively responsible for the operation of the model. They have the expertise and experience in delivering and/or commissioning the program. Two of the co-authors who are representatives of this group provided feedback on the findings of the review. 

Initial searches of the extant Literature revealed few accounts of CI models used to improve access to mental health services. Hence, a review of all CI models in different contexts was undertaken so as to identify findings that were relevant to those for non-acute mental health challenges. 

### 2.1. Identifying the Research Question

Five research questions were identified. They were: What are the rationales for the implementation of CI models in health and health-related services?What are the characteristics of CI models used in different contexts of health and health-related services? What are the challenges and benefits of CI models used in health and health-related services?What are the lessons learned from the Literature on CI models used in health and health-related services?What recommendations can be made for the implementation of CI models for use in non-acute mental health services in different contexts? 

### 2.2. Identifying Relevant Literature

After identifying the research question, PubMed (biomedical Sciences), SCOPUS (multidisciplinary), and Google Scholar (1st 5 pages) databases were searched using the English search terms, ‘centralized intake’ OR ‘central intake’ Or ‘single entry’ with no restriction on dates. Using the same search terms, grey literature including reports and websites of national and international agencies were also searched within the first 5 pages of the Google search engine. The search was undertaken in November 2022. Reference lists of seminal articles were manually searched for additional information. Only reports related to Medicine, Social Sciences, Nursing, and Psychology that had a focus on the model of CI were included in the review. In addition, CI models that were known to exist from the authors’ experience were also included in the review. 

### 2.3. Selection of Studies (Inclusion and Exclusion Criteria)

All articles, reports, and guideline documents were collated in EndNote. After duplicates were removed, titles and abstracts were reviewed for relevance. Reports that did not describe, evaluate, review, or propose practice guidelines for CI models were excluded. 

### 2.4. Collation and Synthesis of Data 

The Literature included in the review was first grouped according to the setting where they were implemented. Next, the data were classified according to the type of literature (descriptions of models; best practice guidelines; reviews and evaluations of models). Key learnings from the data were gleaned from the Literature and described under the following subheadings:Rationale for implementation of CI modelsCharacteristics of CI models Benefits and challenges of these modelsLessons learned from the LiteratureRecommendations for use of a CI model in non-acute mental health services in different contexts

## 3. Results

One hundred and twenty-eight titles were identified that referred to CI. Thirty-four pieces of literature were selected for review, of which 15 sources related to description of the CI model, three related to best practice guidelines, two were reviews, and 14 were evaluations of CI. See Figure 1 for PRISMA flow chat for selection of reports for the review. Most reports were from mental health addiction and counselling services (*n* = 13), substance use services (*n* = 5), and surgical services for arthritis (*n* = 5). Most publications included in this review were from the USA, Canada, and Australia. See Table 1.

### 3.1. Rationale (System Challenges and Objectives) for the Implementation of a CI Model

CI models have been adopted in services for substance abuse [4,5,22,23], mental health [12,20,28,29,32,34,36,37,40], residential care [8,42], cardiac care [25], family support [30], arthritis care/joint replacement [7,13,31,33,44], pediatric rehabilitation [9], and multiple community services [41,43]. The rationale for implementing a CI model varies with each setting and need. Factors that drive the change to a CI model can be classified into those relating to clients, healthcare workers, and the health system. Client factors include increased demand for services [8,13,24,40], difficulty and delays in accessing services [6,7,36,44], lack of knowledge of what services are available [6], having multiple problems and needs [20], and needing to navigate multiple services that operate in siloes [9]. 

Healthcare worker factors that influenced the change to a CI model include frustration and misunderstanding due to a lack of system integration [4], inefficiencies at the level of referral and triage [9,13,25], lack of up-to-date knowledge of the available services for referring clinicians [30], and poor communication among and between service providers [9]. 

System factors include merging of hospitals or health services [12], making better use of the existing service capacity [22], increasing costs [8,25,28,44], the need to integrate multiple service providers [36,37], providing standardized care in a timely and uniform manner [22,25], and the need to improve client outcomes [22]. The rationales (system challenges and the resulting objectives) of the CI model are described in Appendix A.

### 3.2. Characteristics of CI Models

CI models described in the Literature typically have three main elements: initial engagement, screening or assessment, and referral. However, different models utilize these elements either as disparate components or in various combinations as described in Appendix A. 

Two main types of CI models have been reported.

Intake followed by referral to the service providerIntake followed by screening or assessment and referral to service provider

#### 3.2.1. Intake and Referral

In this type of model, central intake is usually via a toll-free telephone line that may run either 24/7 [42] or during office hours [40,43]. The lines are operated by central intake workers who refer callers to the appropriate service based on client need. This model is commonly used when the type of service provider required is clear and clients do not need specialized screening or assessment. Intake workers also provide information about services and programs. This model has been used in homelessness help [42] and community health programs [41,43].

#### 3.2.2. Intake, Screening or Assessment and Referral

Intake followed by screening or assessment is more common and widely utilized. Intake, assessment, and referral can be conducted by different teams or there could be two teams where one conducts the intake and the other conducts the assessment and referral. When intake is conducted separately, intake workers are usually clerical staff who gather preliminary data and schedule further assessments with clinicians or other specialists. This model has been utilized in cardiac services [25], residential care services [8,24] and substance use programs [4]. CI has also been used alongside regular intake [23] to reduce the burden on the latter.

When intake is combined with assessment and referral, intake workers are typically licensed clinical staff such as nurses [28]. In models where assessment or screening is conducted using specified criteria, trained non-clinical staff can conduct screening and referral to the appropriate service provider [7,36]. In a CI model for children’s mental health services, the centralized mobile intake team was staffed by mental health, social service, probation, and school personnel who completed a comprehensive assessment. The intake team also established a service package that comprehensively addressed the child’s needs, which could involve one or multiple agencies [20]. Melathopolous and Cawthorpe [12] describe a model in which telephone intake screening is completed by the mental health clinician and reviewed for urgency. Models that incorporate intake, screening or assessment, and referral have been reported for arthritis services [7,31,44], non-acute mental health services [34,36,40], and substance use programs [22]. CI models which combined initial engagement with screening or assessment and referral typically included complex mechanisms, data management systems, and multidisciplinary teams.

### 3.3. Benefits and Challenges of These Models

Benefits and challenges of CI models are outlined below with further details given in Appendix A.

#### 3.3.1. Intake and Referral

Only one source described the benefits of CI models that involve intake and referral. They included lower call wait times, improved call answer rates, and the ability to respond to higher call volumes [42].

#### 3.3.2. Intake, Screening or Assessment, and Referral

Where available, benefits and challenges are described for clients, service providers, and service systems for specific types of programs. 

##### Substance Abuse Programs

Benefits reported for clients included improved awareness of where to seek help for a drug abuse problem [4], first time engagement with those with a disability or those involved with the justice system [23], reduction in wait time to enter services, and a simplified appointment process [28]. Challenges faced by clients when intake and service delivery were conducted by different organizations included a reduction in service access where clients lost motivation to travel to another site due to increased waiting time, embarrassment, denial, etc. [21]. Berends and Hunter [5] reported improved assessments and increased levels of client satisfaction but not treatment matching. 

Benefits reported for service providers included having realistic expectations from each other and referring consultants [4] and health providers sometimes referring their own clients to CI when they needed additional treatment or were found to be inappropriate for their service [23]. Challenges were not specifically reported.

Benefits for the health system included consistency of approach and an understanding of which programs worked best for which clients [4], savings of up to USD 70 million due to avoiding inappropriate use of emergency departments and state hospitals and savings in operating costs, and the continuing production of performance data that supported system improvement [28]. Challenges for the health system included consistently high demand resulting in insufficient treatment slots and long waiting lists [23].

##### Mental Health Services

Benefits reported for clients included improved access due to simplification of processes of identifying clients’ needs and providing better care [20], as well as reduction in wait times and length of stay [12]. Qualitative reports of client voices have also shown that when care was client-driven, it enabled them to feel valued and have better social life and physical health [6,34]. Challenges included poor outcomes due to difficulty engaging with clients who experienced complex problems such as those involving child protection [6].

Benefits for service providers included a more comprehensive and integrated assessment and time savings [20], a team approach to patient care, better understanding of clients’ history and needs, and avoidance of service duplication [6,9]. Challenges included the need for repeated initial meetings and discussions with local mental health services to allay fears and suspicions of clinicians. In addition, care coordinators who did not have knowledge of local services and agencies found it difficult to make appropriate referrals [38]. 

System benefits included the prevention of the previously occurring disconnection and inefficient determination of the child’s needs and better care [20], improved quality of data [7], and improved service capacity [12]. Challenges included initial difficulties in getting services to work together and later with insufficient mental health specialists in the region [20]. Melathopolous and Cawthorpe [12] reported that, although utilization rates improved marginally, the unmet needs of children and youth did not change. In a study of physicians’ knowledge and attitudes towards CI in child and youth mental health services, Cloutier, Cappelli, Glennie, Charron, and Thatte [27] reported that respondents had concerns about waiting times, availability of services, and lack of feedback from mental health services.

##### Surgery for Arthritis

Benefits for clients included reduced wait times and patients retaining their availability to choose their surgeon [7], improving precision of screening for urgent patients, and sorting of patients according to assessed needs [44]. Challenges were not specifically reported.

Benefits for service providers included increased referral volumes to the next available surgeon [7], and reduced waiting times [35]. Challenges included slow uptake of the service and standards due to slow dissemination and awareness, workload increases, and confusion around stakeholder expectations [7]. 

System benefits included streamlined processes and improved measurement and monitoring of outcomes [7]. System challenges included developing bottlenecks owing to a lack of surgeons and lack of clinicians to undertake screening for non-urgent patients [31] and the need to combine a prioritization strategy with a sorting policy to allow the provision of care for less urgent patients while ensuring the urgent patients do not wait longer to receive theirs [44]. 

##### Home Health Integrated Delivery System

Hamm and Callahan [8] reported increased client satisfaction but also long waiting times and frequent busy signals during peak call times. Benefits for service providers were not specifically reported. The service system benefits of CI included cost effectiveness, better clinical decision making, more appropriate referrals, and better processes and quality outcomes. The CI model of care eventually had to be reverted to the original model due to unexpected changes and a narrowing of the focus of the new system [8].

##### Residential Care Service

Mohr and Bourne [24] reported that the model was well received by all despite conflicting priorities and opinions on staffing. The system developed a consistent and standardized approach to services that targeted client needs and prioritized urgent need. A system challenge was that the skills and experience needed for the intake nurse was not adequately reflected in the classification and Indigenous Elder services had to withdraw due to competing pressures and the need for increased funding for longer hours.

##### Cardiac Care

Bungard, Smigorowsky, Lalonde, Hogan, Maier, and Archer [25] report that the Cardiac EASE (Ensuring Access and Speedy Evaluation) program was investigator-initiated but soon became an operational program funded by Capital Health, thereby highlighting the program’s success.

##### Pediatric Rehabilitation Services

Wittmeier, Restall, Mulder, Dufault, Paterson, Thiessen, and Lix [9] report that clients experienced more equitable wait times based on priority of need and clear and simpler processes for accessing the right services. The authors also report improved communication between therapists, reduced duplication of services, and more accurate wait time data. 

##### Breast Cancer Surgery

Cha, McKevitt, Pao, Dingee, Bazzarelli, and Warburton [39] found that CI for breast cancer surgical referrals reduced wait time from 47 to 41 days despite a slight reduction in operating room availability.

##### Primary care referral to specialists

Milakovic, Corrado, Tadrous, Nguyen, Vuong, and Ivers [3] undertook a review of studies and found that of the 10 studies included in analysis, all reported an absolute reduction in waiting time from initial outpatient visit to a specialist surgeon and internal medicine physician after implementation of the single-entry model. Patient and provider satisfaction with the single-entry model was high in all studies. However, all studies were reported to have bias.

## 4. Lessons Learned from the Literature

Lessons learned from the Literature are listed under the headings: rationale for the use of CI, types of CI models and appropriate contexts for their use, prerequisites for the establishment of a CI model, and mode of intake and possible challenges to overcome. The Literature suggests that CI in healthcare can be defined as a service model that streamlines client intake, assessment, and referral to the right service or healthcare professional in order to adequately utilize available services and facilitate client access. 

### 4.1. Rationale for the Use of CI

Central intake is considered when:There is increased demand for services [8,24,40]Referring physicians and clients are not aware of the available services [30]There are difficulties and delays in accessing services (increased waiting times) [7,22,44]Access to community services needs to be streamlined [41,43]Clients have multiple needs requiring them to navigate multiple services that operate individually [6,8,20,24,36]Multiple providers offer similar services [28] Inefficiencies exist at the level of referral and triage [13,25]Service providers do not communicate with each other [5,9]Hospitals or health services are merged or need to be integrated [12]Existing service capacity needs to be better utilized [23,37] Costs need to be reduced [8]Client outcomes need to be improved [22]

### 4.2. Types of CI Models and Appropriate Contexts for Their Use

#### 4.2.1. Intake and Referral Models of CI Are Better Suited When

The type of service provider required is clear and clients do not need specialized screening or assessment, such as in community support services [41,42,43]. There are multiple and an adequate number of service providers for the number of users [42].

#### 4.2.2. Intake, Screening or Assessment, and Referral Models Are Better Suited When

Clients need to be assessed to determine what service is appropriate for them, such as those clients with multiple needs [20,36]. Clients need to be screened to check eligibility for the service offered, such as triaging clients for cardiac care [25] and surgery for arthritis [44].

### 4.3. Prerequisites for the Establishment of a CI Model

Establishing CI models requires strong individual and organizational leadership [9].The intake system for an area must match community needs and priorities [29].Ensure there are adequate resources both at the CI unit as well as at the referral endpoints [5].CI will invariably increase help-seeking, particularly from new clients who would otherwise not access services. If supply does not meet demand, client frustration and disappointment is likely [23].There needs to be financial commitment from the government (or funder) [9].It is important to have a central intake coordinator when the CI model includes multiple intake workers as well as screening and assessment before referral [9].All stakeholders must be engaged from the outset to garner their support [5,7,9].Ongoing collaboration with stakeholders is crucial for success and quality improvement [5,7].It is important to have standards and accompanying indicators (these are useful for evaluation and future planning and potential modifications to practice) [29].Intake workers should be adequately trained and supported [38].

### 4.4. Mode of Intake 

The telephone is the most common mode of intake in CI models [6,8,12,28,34,36,38,40,42,43], although other modes such as face-to-face or walk-ins, [20,21,33] email [41], fax [8,12], and online referral forms [43] have also been used in conjunction. Face-to-face or walk-ins are typically used as an adjunct to phone intake in certain situations. For instance, when the service targets those who are homeless [42] or have issues related to substance abuse [21], walk-ins and face-to-face intake might be the preferred option. When implementing CI models, consideration must be given to the different possible types of referrals and the preferences of the population groups being targeted. For instance, clients and their family carers may prefer telephone as the mode of referral, whereas General Practitioners and other health professionals may prefer emails or other modes of referral that they commonly utilize. Therefore, when referrals are open to all, multiple modes of intake are preferable.

### 4.5. Possible Challenges to Overcome

When CI is considered for substance abuse programs in resource-poor settings where intake is conducted face-to-face, rather than by telephone or online, and service delivery is conducted by organizations which are at a distance from the intake center, clients may lose motivation and drop out [21].Treatment matching in the case of substance abuse programs continues to be a challenge [5].Consistent high demand for services can result in insufficient treatment slots or service providers and long waiting lists [23].Allaying fears and suspicions and getting services to work together might be challenging [38].Although CI may improve access, it may not change client outcomes (which require a comprehensive and integrated service network) [5].Although CI has control over response time for its initial service, it does not control wait times for the actual mental health services [27].Mental health services are typically overloaded and therefore may not have the time to provide feedback to referring doctors [27].For intake, assessment, and referral models of CI to be made routine, the skills and experience needed for the assessor in the intake team (such as the intake nurse) may not be reflected in the workforce classification [24].CI models for Indigenous services require further research [24].

## 5. Recommendations for a CI Model for Non-Acute Mental Health Services

### 5.1. Planning and Preparation

When planning a CI model for non-acute mental health services, objectives and goals of the program must be defined clearly [7]. The purpose of CI is to improve access to services and as people come to know about it, the demand is likely to increase exponentially to include those who previously encountered several barriers to access services. These individuals are likely to have disabilities or complex problems [23]. Furthermore, services might be sought for children, older people, those in crises, and those who are suicidal. It is the responsibility of the service to have plans to provide the necessary support for all those who access the service [26]. Models need to be flexible enough to adapt to local circumstances [32]. Quality indicators established at the outset will be useful for evaluation, future planning, and potential modifications to practice [29]. Using client outcomes as a measure of success is not realistic given the range of experiences encountered between intake and exit from treatment [5]. More reasonable measures could relate to access, intake, assessment, and referral. Tools and questionnaires should be standardized and easy to use and follow by clients and service providers [7]. Data ownership, oversight, and management must be centralized and decided at the outset [7].

### 5.2. Elements of the Service Model

#### 5.2.1. Initial Engagement

The entry point of the service (intake) must be visible and widely publicized [5,26]. Initial engagement is a crucial part of intake and requires qualified and experienced individuals. Intake must be able to take inquiries from people with related issues, such as sexual assault and emotional, psychological, or social crises, and assist them in a flexible, compassionate, accurate, and quick manner (No wrong door) [5].

Initial engagement is especially important because identifying barriers, collaborative problem solving, and alliance building are important for achieving positive outcomes [29]. In the case of certain programs such as those for substance abuse, matching client preferences to service options should happen from the first contact [29]. In order to prevent disappointment later on, clients must be informed at initial engagement as to what they can and cannot expect from the service.

#### 5.2.2. Screening or Assessment

Assessment must be comprehensive and systematic and must utilize standardized instruments [5]. Clients prefer to receive an immediate and meaningful response following assessment with little separation from service delivery.

#### 5.2.3. Referral

Client needs must be the preferred basis for allocation and prioritization unless proved otherwise [20,29,44]. Information about available services should be both detailed and clear. 

### 5.3. Setting up the Service Network

A CI model is not useful unless it is built on a strong network of integrated services. All stakeholders must be made aware of the purpose and significance of the model as well as their roles and responsibilities [7]. Information, guidelines, and outcomes must be made transparent to all stakeholders [7]. Adequate time and resources will be necessary to ensure that the service is known and recognized in the region. Workforce requirements are likely to grow over time and therefore need to be planned for. Staff must be trained and supported [38]. Clear and regular communication between management and service providers will facilitate better collaboration and engagement. Interprofessional teams have been shown to produce better outcomes for clients [7]. Being flexible and adaptable will foster better engagement and outcomes with service providers and clients.

## 6. Gaps in the Literature and Further Research

Irrespective of whether they are implemented in the private or public sector, the main elements of CI models and their associated challenges appear to be similar. This holds true for rural and urban settings as well. The Literature is limited in regard to the use of CI models specifically for Indigenous and other culturally and linguistically diverse populations. Future research endeavors could focus on identifying characteristics of those who access services for the first time with the introduction of CI. Such studies will inform an understanding of the barriers faced by the more disadvantaged members of society in accessing health and community services. 

## 7. Discussion

This review synthesized the extant Literature on CI models in health and health-related services to describe the various rationales for their use, their characteristics, challenges, and benefits, as well as lessons learned from their implementation. The findings informed recommendations for their implementation in non-acute mental health services. The review also enabled the development of a working definition of a CI model when utilized in health and health-related services.

CI or single-entry models have been developed to address both client and service system concerns. Long waiting times result in deterioration of client heath and loss of confidence in the system. Uneven utilization of services and poor collaboration between agencies or professionals results in a shortfall of cost effectiveness. Apart from bridging the gap between clients and the services they are after, CI can also prevent duplication of care and ensure those who need urgent care are prioritized [45]. However, there are also challenges, such as the initial cost of establishing the CI unit with its associated workforce as well as managing referrals of those who are very unwell [45]. 

The use of CI models at the population level to improve access to care for those with non-acute mental health challenges is relatively new. Hence, such models will need to undergo further evaluation to enable wider application. The growing incidence of public health emergencies such as the COVID-19 pandemic that result in rapid increases in need for mental health services enhances the need for such research. To that end, there are a few lessons that this review offers to policy makers who are considering setting up a CI model for non-acute mental health services. 

For instance, those experiencing emotional of psychological distress who are not keen to spend long hours at the emergency department might decide to access the CI service and it is the responsibility of the service to provide meaningful advice to such individuals [26]. CI models could therefore benefit from having multidisciplinary teams that include social and allied health workers [7]. In addition, the waiting time for already overloaded mental health services is not affected by the introduction of CI [27]. Although CI may improve access, it may not change client outcomes due to a series of client, treatment, and environmental factors [5]. Therefore, client outcomes may not necessarily be a good indicator of success for a CI model [5].

By and large, a CI model works best when it serves as a gateway to a comprehensive network of services that collaborate with each other and with the CI team to ensure that clients receive the most appropriate care, as promptly as possible, in a cost-effective way.

Further research is needed on the use of CI models specifically for Indigenous and other culturally and linguistically diverse populations as well as for clients with substance abuse and dual diagnosis problems.

This review has some limitations. It is very likely that there are examples of CI that are not referred to in this review. Service systems that use the elements of CI may not refer to it as such and hence were not identified in the search. In addition, more details of the systems that existed prior to the implementation of the CI model would have provided a clearer picture of how the system was transformed. This was beyond the scope of this review.

## 8. Conclusions 

The focus of a CI model is to streamline and improve service access for clients. When successfully implemented, CI has been shown to improve access and increase demand for services. However, if CI is not supported by a network of service providers who offer care that is acceptable to clients, the purpose of its implementation could be lost. This review is not a comprehensive treatise on CI models. It does, however, provide an overview on the topic with some guidelines for its use. 

## Figures and Tables

**Figure 1 ijerph-20-05747-f001:**
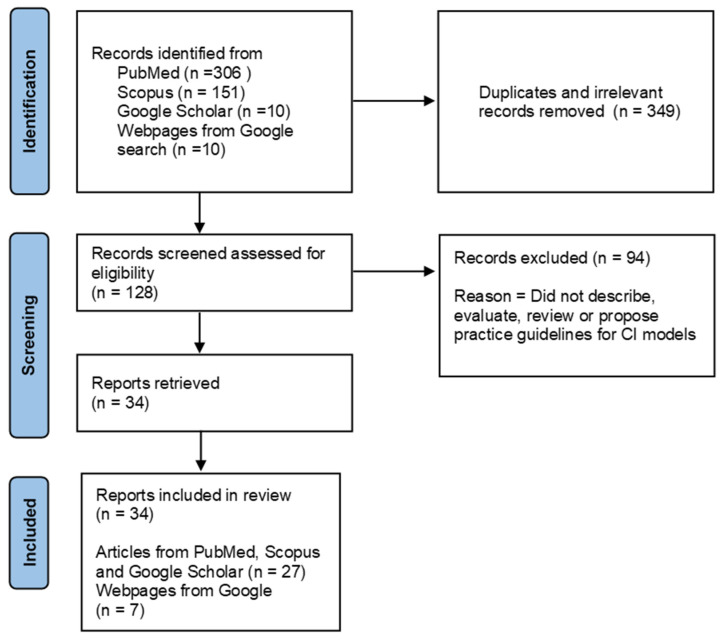
PRISMA flow Chart to show selection of reports for the scoping review.

**Table 1 ijerph-20-05747-t001:** Summary of articles on CI selected for the review.

No.	Author, (Ref)	Country	Setting	Type of Article
1	Levie, Claxton and Barnes [4]	USA	Drug abuse	Description
2	Sawyer and Moreines [20]	USA	Rural children’s mental health services	Description
3	Rohrer, et al. [21]	USA	Costs in substance abuse programs	Evaluation
4	Hamm and Callahan [8]	USA	Care management within a home health integrated delivery system	Description
5	Scott, et al. [22]	USA	Substance abuse	Evaluation
6	Woods, et al. [23]	USA	Substance abuse program	Description
7	Mohr and Bourne [24]	Canada	Community healthcare programs	Description
8	Bungard, et al. [25]	Canada	Improving access to elective cardiac consultations	Description
9	Duncombe [26]	Australia	Adult counselling in rural community health	Review
10	Berends and Hunter [5]	Australia	Alcohol and drug systems	Evaluation
11	Cloutier et al. [27]	Canada	Child and youth mental health service	Evaluation
12	Schuble, et al. [28]	USA	Mental health and addictions services	Evaluation
13	Ontario centre of excellence for child and youth mental health [29]	Canada	Child and youth mental health	Best practice guidelines
14	Early Childhood Iowa Quality Services and Programs Component group [30]	USA	Early childhood services (family support)	Best practice guidelines
15	Barber, Patel, Woodhouse, Smith, Weiss, Homik, LeClercq, Mosher, Christiansen, Howden, Wasylak, Greenwood-Lee, Emrick, Suter, Kathol, Khodyakov, Grant, Campbell-Scherer, Phillips, Hendricks and Marshall [13]	Canada	Osteoarthritis and rheumatoid arthritis	Best practice guidelines
16	Suter, et al. [31]	Canada	Hip and knee replacement surgery for arthritis	Evaluation
17	Rush and Saini [32]	Canada	Mental health and addiction services	Review
18	Wittmeier, Restall, Mulder, Dufault, Paterson, Thiessen and Lix [9]	Canada	Pediatric physiotherapy	Evaluation
19	Dembo, et al. [33]	USA	At risk youth in the justice system	Evaluation
20	Waks, et al. [34]	Australia	Partners in Recovery for severe mental illness	Evaluation
21	Isaacs, Sutton, Dalziel and Maybery [6]	Australia	Partners in Recovery for severe mental illness	Description
22	Damani, MacKean, Bohm, Noseworthy, Wang, DeMone, Wright and Marshall [7]	Canada	Hip and knee replacement surgery for arthritis	Evaluation
23	Damani, et al. [35]	Canada	Hip and knee replacement surgery for arthritis	Evaluation
24	Isaacs, et al. [36]	Australia	Partners in Recovery for severe mental illness	Description
25	Melathopolous and Cawthorpe [12]	Canada	Child and Adolescent Mental Health and Psychiatry Program	Description
26	Hutt-MacLeod, et al. [37]	Canada	First Nations Youth mental healthcare service	Description
27	Isaacs and Firdous [38]	Australia	Partners in Recovery for severe mental illness	Description
28	Cha, et al. [39]	Canada	Breast cancer surgery	Evaluation
29	Northwestern Melbourne Primary Health Network [40]	Australia	Sub-acute mental health problems	Description
30	New South Wales Health. Southern NSW Local Health District [41]	Australia	Local health District Community Health Central Intake Service	Description
31	Milakovic, et al. [3]	Canada	Outpatient visits to specialist physicians and allied health professionals	Evaluation
32	City of Toronto [42]	Canada	Homeless help	Description
33	Grampians Health Ballarat [43]	Australia	Referral management service for community programs	Description
34	Marshall, et al. [44]	Canada	Hip and knee replacement surgery for arthritis	Evaluation

## Data Availability

Data is contained within the article or Appendix A.

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
