# Peer review of "Centralized Intake Models and Recommendations for Their Use in Non-Acute Mental Health Services: A Scoping Review"

_ijerph, 2023, doi:10.3390/ijerph20095747_

Round 1

Reviewer 1 Report

To the authors,

Thank you for submitting your manuscript. It deals with an interesting issue and conducting a scoping review is a good choice. However, as it stands, the manuscript needs to undergo some revisions in order to be eligible for publication. Some major concerns are related to the scope of the review and what is actually included and the lack of a discussion.

Below I provide some comments and suggestions for you to address in your further work with the presentation of your study.

The abstract has a good structure and is well-written.

I suggest using Scoping review as one of your key words.

The introduction seems to be a bit short, and I am lacking a clear argument for conducting this particular study – other than, that it has not been done before. Please elaborate on why this study is of importance. 

The section on materials and methods has a good structure that to some extent follows the stages of Arksey and O’Malleys framework. It is stated that the specific methodology was adopted to undertake the review. Please elaborate on how and the reasons for doing so?

The Arksey and O’Malley framework includes an optional stage which includes consultation exercise. What was the reason for not including this in this study? I am aware that Arksey and O’Malley describe this as optional, but several of the later sources for conducting a scoping review mentions this as an important aspect of the methodology.

Research questions – you state that four research questions guided the review, however five are listed. Please revise.

Please clarify how these are related to the specific non acute mental health context?

How do you differentiate research question 3, 4 and 5? (challenges and benefits of these models, the lessons learned from the literature on CI models, and recommendations that can be made for the implementation of CI models).

Under 2.2. Identifying relevant literature, you write that ‘Only reports related to Medicine, Social Sciences, Nursing and Psychology that had a focus on the model of CI were included in the review.’ I suggest including this focus in your research questions – perhaps where you currently write: ‘What are the characteristics of CI and referral models used in different contexts of health and human services’. Perhaps reuse the phrase from research question 5 (for use in non-acute mental health services in different contexts) if this is your focus.

2.3. Selection of studies – it would be helpful to include a PRISMA flowchart to illustrate the procedure of selecting/screening of studies. Furthermore, please elaborate on whether you used any inclusion or exclusion criteria as you reviewed the studies for relevance.

Results – please revise the text, so that no sentences start with numbers – also please follow the norms for reporting numbers in text or actual numbers (one to ten is spelled out…).

Please clarify the use of brackets in this section as well. Are these references to literature? If yes, how does the phrase ‘the most reports’ justify the use of only one reference? Or do they refer to the number of studies which report on each topic?

Results (including tables)

Considering that the focus of this scoping review is on ‘non-acute mental health services’ I do not see the relevance of reporting all other settings. Please clarify this choice – I suggest removing any content that is not related to that specific context.

The manuscript needs to include a discussion of the findings.

Reviewer 2 Report

Interesting topic, definitely worth investigation. Nevertheless some issues need to be adressed.

1. Introduction. Introduction is brief and does not provide sufficient background. The Authors state that "There is no single definition for this model of care and it is utilized in various forms depending on the need and nature of the situation". That is true, of course, however as the Authors decide to focus on that topic, some clarification precising what is planed to be analysed should be provided.

2. The Authors focus on central intake regardless of the reffered services profile.  This is however an important issue - intake, but where: to what kind of system? The system profile should be discussed: what kind of network (parallell sites with similar offer; hierarchical structure?) Public? Private? Reimbursed? If the Authors are unable to extract these information from the analyzed papers, it should be at least mentioned in the Introduction and Limitations sections.

The paper discusses the Central Intake understood as a change from the previous way of refferal. It should be explained what kind of refferal system was used earlier - otherwise the information is incomplete.

3. In the Introduction the Authors refer to the United States and in the result section the analyzed papers are from USA, Canada and Australia. This is only a part of the world, and all three countries have satisfying economic development. It should be at least discussed. It would be interesting to know how the refferal system works in other regions of the globe.

4. The Table are very long. I would propose to prepare more concise summary for the main manuscript and transfer detailed Tables to the suplementary materials.

Round 2

Reviewer 1 Report

Thank you for this revised version of your manuscript and your response to the suggested revisions. I beleive it has improved significantly. I only have a few remarks:

I suggest making the wording of reqearch questions 2 and 3 more alined - focused on health and health-related services.

The discussion that has been included is short and draws on only one reference. I would expect that you have more to dicsuss based on the 34 studies on CI that you have read and included in this review. I strongly suggest taking another look at this section and elaborate on how CI could be used in the setting you have foced on. 

Reviewer 2 Report

The paper is suitable for publication

Author Response

Thank you for your positive comment.